# Exopolysaccharide Carbohydrate Structure and Biofilm Formation by *Rhizobium leguminosarum* bv. *trifolii* Strains Inhabiting Nodules of *Trifolium*
*repens* Growing on an Old Zn–Pb–Cd-Polluted Waste Heap Area

**DOI:** 10.3390/ijms22062808

**Published:** 2021-03-10

**Authors:** Ewa Oleńska, Wanda Małek, Urszula Kotowska, Jerzy Wydrych, Weronika Polińska, Izabela Swiecicka, Sofie Thijs, Jaco Vangronsveld

**Affiliations:** 1Department of Microbiology and Biotechnology, Faculty of Biology, University of Białystok, 1J Ciołkowski, 15-245 Białystok, Poland; izabelas@uwb.edu.pl; 2Department of Genetics and Microbiology, Institute of Biological Sciences, Faculty of Biology and Biotechnology, Maria Curie-Skłodowska University, 19 Akademicka, 20-033 Lublin, Poland; wanda.malek@poczta.umcs.lublin.pl; 3Division of Environmental Chemistry, Department of Analytic and Inorganic Chemistry, Faculty of Chemistry, University of Białystok, 1K Ciołkowski, 15-245 Białystok, Poland; ukrajew@uwb.edu.pl; 4Department of Functional Anatomy and Cytobiology, Institute of Biological Sciences, Faculty of Biology and Biotechnology, Maria Curie-Skłodowska University, 19 Akademicka, 20-033 Lublin, Poland; jerzy.wydrych@poczta.umcs.lublin.pl; 5Doctoral School of Exact and Natural Sciences, University of Białystok, 1K Ciołkowski, 15-245 Białystok, Poland; w.polinska@uwb.edu.pl; 6Laboratory of Applied Microbiology, Faculty of Biology, University of Białystok, 1J Ciołkowski, 15-245 Białystok, Poland; 7Centre for Environmental Sciences, Faculty of Sciences, Hasselt University, Agoralaan D, B-3590 Diepenbeek, Belgium; sofie.thijs@uhasselt.be (S.T.); jaco.vangronsveld@uhasselt.be (J.V.); 8Department of Plant Physiology and Biophysics, Institute of Biological Sciences, Faculty of Biology and Biotechnology, Maria Curie-Skłodowska University, 19 Akademicka, 20-033 Lublin, Poland

**Keywords:** exopolysaccharide (EPS), rhizobia, heavy metals, biofilm, gas chromatography-mass spectrometry (GC-MS), retention index, confocal laser scanning microscopy (CLSM), nucleic acid staining dyes, adaptation to toxic metals

## Abstract

Heavy metals polluting the 100-year-old waste heap in Bolesław (Poland) are acting as a natural selection factor and may contribute to adaptations of organisms living in this area, including *Trifolium repens* and its root nodule microsymbionts—rhizobia. Exopolysaccharides (EPS), exuded extracellularly and associated with bacterial cell walls, possess variable structures depending on environmental conditions; they can bind metals and are involved in biofilm formation. In order to examine the effects of long-term exposure to metal pollution on EPS structure and biofilm formation of rhizobia, *Rhizobium leguminosarum* bv. *trifolii* strains originating from the waste heap area and a non-polluted reference site were investigated for the characteristics of the sugar fraction of their EPS using gas chromatography mass-spectrometry and also for biofilm formation and structural characteristics using confocal laser scanning microscopy under control conditions as well as when exposed to toxic concentrations of zinc, lead, and cadmium. Significant differences in EPS structure, biofilm thickness, and ratio of living/dead bacteria in the biofilm were found between strains originating from the waste heap and from the reference site, both without exposure to metals and under metal exposure. Received results indicate that studied rhizobia can be assumed as potentially useful in remediation processes.

## 1. Introduction

Exopolysaccharides (EPS) represent a group of polysaccharides synthesized and secreted to the external environment or synthesized extracellularly by cell wall-associated enzymes of many Gram-positive and Gram-negative bacteria [1,2,3,4]. Exopolysaccharides consist of repeated units of sugar monomers and some non-carbohydrate substituents, i.e., phosphates, acetyls, succinate, glycerol, or pyruvate [5,6]. These slime polymers possess a heterogenous structure, which determines their unique properties and functions, highly dependent on the peculiar conditions of niches inhabited by their host [2]. Major functions of EPS involve the protection of cells from adverse environmental conditions, storage of nutrients, and assistance in formation of mono- or multispecies microconsortia, i.e., biofilms or biogranules [7,8,9,10]. Exopolysaccharides accumulate on bacterial surfaces, providing protection by stabilization of the membrane against external factors. In nature, many bacteria usually do not occur as single-celled planktonic organisms but exist in highly organized multi-cellular associations. This is mainly due to the unique properties of the EPS, involving improvement of inter-cellular as well as cell-surface adhesion [11,12,13], where they communicate via quorum sensing (QS) pathways [14]. Such a multi-organismal composition offers to its inhabitants a significantly higher resistance to multiple and changing environmental factors than individual planktonic forms represent [15,16]. It was revealed that bacteria in a biofilm form tolerate a wide range of challenging fluctuations in temperature, humidity, salinity, and pH, as well as the presence of bacteriocins, antibiotics, or toxic metal ions in the environment [17,18].

The quantitative and qualitative composition of EPS varies [2,13]. EPS are composed of a limited number of recurring monosaccharides and their derivatives, representing a complex and variable mosaic of linked monosaccharide units, arranged in linear or branched configurations determining their rheological properties. The rigid EPS backbone is composed by monomers connected predominantly by β–(1,4)- or β–(1,3)-glycoside linkages, while monosaccharides connected by α–(1,2)- and α–(1,6)- glycoside linkages provide a flexible character to these extracellular polymers [19,20]. According to their composition, the EPS are classified as homopolysaccharides when comprised with monosaccharides of a single structural unit, heteropolysaccharides when a mosaic is composed of two or more repeating structural units, and polysaccharides with an irregular structure [2,6]. The abundance of the carbohydrates in the composition of an EPS determines the absorptive character of this polymer towards multiple compounds, including metal ions. Extracellular immobilization of metal ions (biosorption or bioaccumulation) by microbial metabolites is one of the mechanisms of metal resistance in the *Bacteria* domain [21,22,23]. In EPS, the chemically neutral homopolysaccharides and polyanionic heteropolysaccharides rich in uronic acids, i.a., glucuronic, galacturonic, and mannuronic acids as well as pyruvate were detected [24,25]. Sequestration of metal ions into bacterial exo-polymers is due to electrostatic interaction with negatively charged residues, i.e., uronic acids [26]. Inorganic phosphates and, less frequently, sulfates enhance the potential of EPS to immobilize toxic metal ions [27].

Toxic concentrations of metals in soils can act adversely on all edaphic biota, as well may be readily transferred to plants and enter food chains. In order to mitigate the further dispersal of such metals, many physico-chemical remediation methods have been developed, i.e., soil washing, thermal desorption, encapsulation, electrokinetic separation, solidification, precipitation, and ion exchange [28]. However, these techniques were found to considerably affect soil structure and to have adverse or even lethal effects on soil microbiota [29]. To avoid such inadmissible consequences, more environmentally friendly bioremediation technologies have been developed based on the use of living organisms to remediate and restore disrupted sites. For these purposes, adapted microbes and plants are used to develop bio/phytoremediation strategies [30,31,32,33,34,35,36]. Many metals are present in the environment as natural constituents of the lithosphere but also resulting from anthropogenic activities. At higher concentrations, both non-essential but also essential trace elements can become toxic [37]. Toxicity can be due to modifications in the conformation of macromolecules, disturbance of metabolic pathways, induction of oxidative damage and acceleration of senescence, necrosis, and genotoxicity, leading to reductions in numbers of individuals, population size, and, as a result, limitation of a genetic polymorphism and adaptability of populations [38,39,40,41,42,43].

As metals can act negatively on the survival of organisms, including bacteria, they function as natural selective factors shaping the population structure, favoring the fittest individuals and bringing the less adaptive ones to extinction [44,45,46,47]. In Poland, the about 100-year-old postindustrial calamine waste heap of Bolesław, polluted mainly with Zn, Pb, and Cd with total concentrations in soil up to 52,795 µg g^−^^1^, 578 µg g^−^^1^, 605 µg g^−^^1^, respectively [36], is an area where genotypes adapted to such specific and extreme conditions can develop. The Bolesław waste heap is inhabited by a vegetation cover of natural origin that developed as a result of natural succession. On this site, *Trifolium repens* is present, together with its natural symbionts [38]. *T. repens* as a legume (Fabaceae) enters into a mutual symbiotic association with *Rhizobium leguminosarum* bv. *trifolii*, a Gram-negative rhizobial bacterium. Rhizobia represent an ecological group of rhizobacteria, commonly synthesizing slime EPS, and inhabit soils for years as saprophytes entering into symbiosis with specific legume plant species during the vegetative season [48,49]. Considering that rhizobia spend most of their time as free-living inhabitants of soil particles or plant roots surfaces, it was proposed that a biofilm life style should be an ideal form of rhizobia to subsist and increase the probability of nodulation [12,50,51,52]. A symbiotic association of a leguminous plant root with rhizobia is visible as nodules, created by a host-plant that offers a safe niche to its symbionts where also assimilates are delivered; in return, rhizobia fix the diatomic nitrogen gas that is not accessible for plants [49]. As a consequence, this co-evolutionary solidified plant–microbe association may involve adaptive mechanisms, including the synthesis of EPS with a structure specific to waste heap inhabitants. Therefore, in order to determine the diversity of rhizobia from waste heap origin in comparison to these originating from a non-polluted reference site, seven *R. leguminosarum* bv. *trifolii* strains isolated from nodules of *T. repens* growing on the Bolesław waste heap and a non-polluted reference area in Bolestraszyce were grown without exposure as well as with exposure to toxic concentrations of Zn, Pb, and Cd. The strains were investigated for their quantitative and qualitative characteristics of the sugar fraction of the EPS using gas chromatography–mass spectrometry (GC-MS), as well as for biofilm formation and structural characteristics using a colorimetric crystal violet test and confocal laser scanning microscopy (CLSM).

## 2. Results

### 2.1. Exopolysaccharide (EPS) Carbohydrates Analysis

#### 2.1.1. Qualitative Determination of EPS Carbohydrates

In total, 82 EPS samples were analyzed. They were produced by *R. leguminosarum* bv. *trifolii* strains originating from root nodules of *T. repens* from the polluted Bolesław waste heap and the non-polluted reference area in Bolestraszyce. The strains were cultured under non-exposed control conditions (C-group) and metal-exposed conditions in medium with: 0.5 mM Zn (Zn 0.5 group) and 2.5 mM Zn (Zn 2.5 group), 0.5 mM Pb (Pb 0.5 group) and 2.5 mM Pb (Pb 2.5 group), as well as 0.05 mM Cd (Cd 0.05 group) and 1.0 mM Cd (Cd 1.0 group). Thirty different carbohydrate-type compounds were identified (Table 1). The identification of eight compounds was not possible (Appendix A
Appendix A). Twenty-three identified compounds were monosaccharides (i.e., α-D,L-lyxopyranose, α-D-arabinopyranose, D-rhamnose, L-rhamnose monohydrate, levoglucosan, 6-deoxy-D-glucose, methyl α-D-mannopyranoside, galactofuranoside, α-D-mannopyranose, D-fructose, β-D-galactopyranoside, α-D-galactofuranose, α-D-mannofuranose, α-glucofuranose, D-galactose, α-D-glucopyranoside, α-D-glucopyranose, D-glucose, β-D-galactopyranose, β-D-mannopyranose, β-talopyranose, sedoheptulose, and β-D-glucopyranose), four of them were acids (i.e., galactonic acid, gluconic acid, glucuronic acid, and D-glucuronic acid), two were alcohols (i.e., glucitol and *myo*-inositol), and one was an amino-carbohydrate (N-acetyl-D-glucosamine). Most of the identified monosaccharides were cyclic hexoses (37%) (pyranoses and furanoses). D-glucose as well as D-galactose were occurring in the highest (10%) amounts of carbohydrate derivatives identified in this study. The mean retention times (t_Rmean_), calculated (I) and reference (I_REF_) indices of retention, and the mass-to-charge ratio (m/z), which are the basis for identification of the compounds in the EPS of *R. leguminosarum* bv. *trifolii* strains, are presented in Appendix A
Appendix A.

In the case of eight carbohydrate-type compounds, an unambiguous identification was not possible (Appendix A
Appendix A). On the basis of the I values and comparison with literature values (I_REF_), as well as the analysis of often poorly developed spectra, assumptions can be made concerning the structure of these compounds. The compound with retention time 41.414 min and I = 1883 might be α-talofuranose (I_REF_ = 1882) or β-fructopyranose (I_REF_ = 1887); the compound with retention time 42.514 min and I = 1896 might be α-D-fructopyranose (I_REF_ = 1902), α-D-galactopyranose (I_REF_ = 1899), or β-sorbopyranose (I_REF_ = 1900). The compound with retention time 43.847 min (I = 1912) might be γ-glucuronolactone (I_REF_ = 1902) or α-D-fructopyranose (I_REF_ = 1913). Peaks registered at retention times 45.419 min (I = 1930) and 46.006 min (I = 1937) can be δ-tagatose (I_REF_ = 1934) or galacturonic acid (I_REF_ = 1936). The substances with highly comparable retention times 47.280 min (I = 1952), 47.532 min (I = 1955), and 47.798 min (I = 1958) might be carbohydrate acids, i.e., β-D-glucopyranuronic acid (I_REF_ = 1957), gulonic acid (I_REF_ = 1958), or galacturonic acid (*I_REF_* = 1954).

The frequencies (f) of the identified monosaccharides (number of samples with a particular monosaccharide divided by the total number of analyzed samples, multiplied by 100) and the percentage composition (% C) of particular carbohydrates in the analyzed samples (sum of a particular monosaccharide peak area divided by the sum of all registered peak areas, multiplied by 100) in all studied samples are presented in Appendix A
Appendix A. Glucitol (98.78%), α-D-glucopyranoside (84.15%), methyl α-D-mannopyranoside (69.51%), D-fructose (63.41%), β-D-galactopyranose (59.76%), and α-D-mannopyranose (57.32%) occurred at the highest frequencies in the analyzed *R. leguminosarum* bv. *trifolii* EPS. The lowest frequencies (f = 1.22%) could be attributed to α-D,L-lyxopyranose, D-rhamnose, D-glucuronic acid, and myo-inositol. The highest percentages composition of monosaccharides in the analyzed samples were for glucitol (53.86–100%), β-D-galactopyranoside (5–46.04%), α-D-glucopyranoside (0.11–16.49%), and D-galactose (0.05–15.42%); the lowest one was for myo-inositol (0.013%).

#### 2.1.2. Quantitative Analysis of EPS Carbohydrates

Concentrations of monosaccharides in *R*. *leguminosarum* bv. *trifolii* EPS were calculated using monosaccharide standard curves (Appendix A) in relation to the calculated parameters as follows: coefficient of variation (CV), limit of the compound determination (LOD), and limit of the quantification (LOQ) (Appendix A
Appendix A). In general, the highest concentrations of monosaccharides were observed for galactonic acid (4346.712 µg mL^−1^) and glucitol (1816.583 µg mL^−1^); the amino-saccharide N-acetyl-D-glucosamine was detected at concentrations of 17.52–19.86 µg mL^−1^, while lower concentrations of L-rhamnose monohydrate (6.41 µg mL^−1^) and D-rhamnose (6.69 µg mL^−1^) were found (Appendix A). Glucitol, β-D-galactopyranoside, α-D-glucopyranoside, and D-galactose were the main EPS components of *R. leguminosarum* bv. *trifolii* and comprised, respectively, up to 100%, 46%, 16.5%, and 15.42% of their composition. D-rhamnose (6.69 µg mL^−1^), showed the lowest concentration in the analyzed rhizobial EPS (Appendix A
Appendix A).

#### 2.1.3. Quantitative and Qualitative Comparison of Carbohydrate Content between *R. leguminosarum* bv. *trifolii* of Waste Heap and Reference Grassland Origin

Analysis of 82 EPS samples of *R. leguminosarum* bv. *trifolii* isolated from nodules of *T. repens* growing on the Bolesław metal-polluted waste heap area and the non-polluted reference grassland in Bolestraszyce revealed substantial differences in quantity and concentration of carbohydrates (Table 1). All identified carbohydrates produced by rhizobia not exposed to toxic metal concentrations (control group, C), circa 57%, were observed in both EPS of rhizobia from non-polluted reference and those from waste heap origin. Some monosaccharides were specific to one of both groups: for instance, α-D,L-lyxopyranose and galactonic acid were found exclusively in EPS of rhizobia originating from the metal-polluted area, while levoglucosan, α-D-mannopyranose, and gluconic acid were characteristic for *R. leguminosarum* bv. *trifolii* from the non-polluted reference area. Figure 1 presents quantitative and qualitative differences of carbohydrates in the EPS produced by *R. leguminosarum* bv. *trifolii* originating from the Bolesław waste heap and the non-polluted reference site in Bolestraszyce.

Substantially lower percentages (36.7% for rhizobia of reference area origin and 26.7% for rhizobia of waste heap origin) of carbohydrates were detected in the EPS of rhizobia exposed to 0.05 mM Cd in comparison with non-exposed ones. Compared to the latter, the EPS of the 0.05 mM Cd-exposed ones did not contain α-D,L-lyxopyranose, 6-deoxy-D-glucose, methyl α-D-mannopyranoside, D-fructose, α-D-glucopyranose, β-D-galactopyranose, sedoheptulose, and galactonic acid. Other carbohydrates were only identified in the Cd-exposed rhizobia but not in the non-exposed ones. Substantial reductions in concentrations of D-galactose, D-glucose, and glucitol were observed in the EPS of the 0.05 mM Cd-exposed in comparison to non-exposed rhizobia. No significant quantitative differences were observed between carbohydrates in the EPS of 0.05 mM Cd-exposed rhizobia of both reference and waste heap origin, but, in contrast to strains from the reference area, the EPS from rhizobia of waste heap origin were deficient in galactofuranoside, α-glucofuranose, and D-galactose.

Exopolysaccharides of rhizobia exposed to 1.0 mM Cd contained substantially lower percentages of carbohydrates (respectively, 30% and 23% for rhizobia from the non-polluted reference and the polluted site) than non-exposed rhizobia; qualitative and quantitative differences were observed in comparison to the 0.05 mM Cd-exposed rhizobia. Compared to the non-exposed rhizobia, significant increases were detected in the concentrations of methyl α-D-mannopyranoside, sedoheptulose, and glucitol in the EPS of 1.0 mM Cd-exposed ones. The 1.0 mM Cd-exposed rhizobia further did not contain detectable concentrations of levoglucosan, 6-deoxy-D-glucose, galactofuranoside, α-glucofuranose, D-galactose, α-D-glucopyranose, D-glucose, β-D-glucopyranose, galactonic, and gluconic acid concentrations. Moreover, in the EPS of 1.0 mM Cd-exposed rhizobia, the monosaccharides α-D-arabinopyranose and β-talopyranose were detected. Within the 1.0 mM Cd-exposed rhizobia, quality differences in carbohydrates content were observed between strains originating from the non-polluted reference area and the ones from the metal-polluted waste heap: in EPS of rhizobia of waste heap origin, β-talopyranose was identified, while α-D-arabinopyranose, D-fructose, and β-D-galactopyranoside were detected in EPS of rhizobia from the reference area.

Exopolysaccharides of rhizobia exposed to 0.5 mM Zn contained, respectively, 70% and 50% (for rhizobia from the non-polluted reference and the polluted site) of all carbohydrates identified in this study. In comparison to non-exposed rhizobia, different monosaccharide units were identified in EPS of 0.5 mM Zn-exposed rhizobia, i.e., α-D-arabinopyranose, L-rhamnose monohydrate, α-D-galactofuranose, α-D-mannofuranose, β-D-mannopyranose, β-talopyranose, glucuronic acid, D-glucuronic acid, and N-acetyl-D-glucosamine. Levoglucosan, 6-deoxy-D-glucose, α-D-glucopyranose, and β-D-glucopyranose were not detected in EPS of the 0.5 mM Zn-exposed rhizobia, but they were present in EPS of non-exposed rhizobia. In the EPS of *R. leguminosarum* bv. *trifolii* exposed to 0.5 mM Zn, significantly higher concentrations of β-D-galactopyranoside and glucitol, as well as significantly lower concentrations of D-fructose, D-galactose, and D-glucose, were detected in comparison with non-exposed strains. The 0.5 mM Zn intragroup qualitative and quantitative comparison revealed substantial differences between the carbohydrates concentration of EPS from rhizobia originating from the non-polluted reference site and the metal-polluted site. Significantly lower concentrations of β-D-galactopyranoside and gluconic acid were found in EPS of rhizobia from waste heap origin in comparison with those from the non-polluted reference area. Moreover, after exposure to 0.5 mM Zn, α-D-arabinopyranose, L-rhamnose monohydrate, galactofuranoside, α-D-galactofuranose, α-D-mannofuranose, D-galactose, D-glucuronic acid, and N-acetyl-D-glucosamine were detected only in EPS of rhizobia from reference origin. Sedoheptulose was found exclusively in EPS of 0.5 mM Zn-exposed rhizobia from waste heap origin.

Exopolysaccharides of rhizobia exposed to 2.5 mM Zn contained 13% of the carbohydrates identified in this study. Aside from carbohydrates identified in EPS of the non-exposed and 0.5 mM Zn-exposed rhizobia, α-D-glucopyranoside was detected exclusively in rhizobia originating from the non-polluted reference area, while sedoheptulose was found in the EPS of rhizobia of waste heap origin. Glucitol was found in the highest concentrations in EPS of 2.5 mM Zn-exposed rhizobia, similar to values observed in EPS of 0.5 mM Zn-exposed ones.

In the EPS of *R. leguminosarum* bv. *trifolii* exposed to 0.5 mM Pb, respectively 70% and 57% of total carbohydrates identified in this study were found in strains originating from the non-polluted reference area and waste heap origin. In contrast to the non-exposed control, the following carbohydrates were identified in the EPS of 0.5 mM Pb-exposed rhizobia: α-D-arabinopyranose, D-rhamnose, L-rhamnose monohydrate, α-D-galactofuranose, α-D-mannofuranose, β-D-mannopyranose, β-talopyranose, glucuronic acid, *myo*-inositol, and N-acetyl-D-glucosamine. In EPS of 0.5 mM Pb-exposed rhizobia, significantly higher concentrations of methyl α-D-mannopyranoside, β-D-galactopyranoside, α-D-glucopyranoside, β-D-galactopyranose, and glucitol were detected, whereas concentrations of D-fructose and D-galactose were lower in comparison with the non-exposed ones. Comparison of the intragroup Pb 0.05 reference and waste heap rhizobia EPS contents revealed a significantly higher concentration of D-glucose and galactonic acid in rhizobia from Bolesław in comparison to those from the reference area. Moreover, D-rhamnose, α-D-mannofuranose, and α-glucofuranose appeared as specific to EPS of rhizobia from waste heap origin, while L-rhamnose monohydrate, galactofuranoside, α-D-galactofuranose, sedoheptulose, gluconic acid, myo-inositol, and N-acetyl-D-glucosamine were characteristic to EPS of reference area rhizobia.

After exposure to the higher dose of 2.5 mM Pb, *R. leguminosarum* bv. *trifolii* EPS comprised respectively 27% and 23% (for rhizobia from the polluted and reference site) of total carbohydrates identified in the study. Methyl α-D-mannopyranoside, α-D-mannopyranose, and glucitol were found as common carbohydrates for 2.5 mM Pb-exposed as well as non-exposed rhizobia. Glucitol was found in EPS of 2.5 mM Pb-exposed rhizobia as the compound with the highest concentration among others detected in this group and the control group carbohydrates and of a similar quantity to EPS of Pb 0.5 group rhizobia. Sedoheptulose was found as a specific carbohydrate to EPS of rhizobia originating from the reference Bolestraszyce grassland, while β-D-galactopyranoside and β-talopyranose were specific to the waste heap Bolesław rhizobia EPS.

#### 2.1.4. Quantitative and Qualitative Comparison of Carbohydrate Content between Strains of *R. leguminosarum* bv. *trifolii* Originating from a Waste Heap and a Non-Polluted Reference Area

In order to identify the *R. leguminosarum* bv. *trifolii* strain equipped with the widest range of EPS carbohydrates as a basis for potential use in bioremediation, the quantitative as well as qualitative content of EPS of three *R. leguminosarum* bv. *trifolii* strains isolated from root nodules of *T. repens* growing on the waste heap of Bolesław (4.1H, 5.2H, 7.2H) and four strains from a non-polluted reference grassland in Bolestraszyce (3.9K, 4.4K, 5.10K, 8.8K) not exposed (control) and exposed to the highest metal concentrations, i.e., 1.0 mM Cd, 2.5 mM Zn, and 2.5 mM Pb, were compared. Among the 30 carbohydrates that were identified in total, seven were detected in bacteria exposed to the highest doses of metals, i.e., methyl α-D-mannopyranoside, α-D-mannopyranose, D-fructose, α-D-glucopyranoside, β-D-galactopyranose, sedoheptulose, and glucitol (Appendix A
Appendix A). Levoglucosan, galactofuranoside, β-D-galactopyranoside, α-glucofuranose, D- galactose, D-glucose, α-D-glucopyranose, β-D-glucopyranose, and gluconic acid were detected only in EPS of non-exposed rhizobia, while β-talopyranose was found as exclusive to waste heap origin rhizobia under Cd exposure. Therefore, *R. leguminosarum* bv. *trifolii* strain 7.2 H as well as strains 4.1 H and 5.2 H can be suggested as promising in cases of Cd stress tolerance. Moreover, sedoheptulose, found as a constituent of EPS in rhizobia originating from the non-polluted reference area, and the waste heap exclusively under non-exposed and Cd-exposed conditions showed a significantly higher percentage content under metal exposure in comparison to non-exposed conditions. By consequence, all studied strains can be considered as potential tools equipped in sedoheptulose interacting towards Cd stress. Moreover, the glucitol concentration was significantly higher under Cd, Zn, and Pb exposure in comparison to EPS of rhizobia not exposed to metals.

### 2.2. Biofilms Structure Analysis Using CLSM

All *R. leguminosarum* bv. *trifolii* strains cultured under non-exposed as well as Zn, Pb, and Cd-exposed conditions analyzed using the colorimetric crystal violet method showed an absorbance (OD_490_) in the range of 0.30–0.85, and, according to Christensen et al. [53], this indicates an ability for biofilm formation (data not presented). CLSM analysis of biofilms indicated significant differences in mean thickness of the biofilms formed by rhizobia originating from the non-polluted reference ($\bar{x}$_K_) and the polluted waste heap area ($\bar{x}$_H_) not exposed to metals (control group, C) (*p* = 0.0015) and exposed to Cd (Cd 0.05 group) (*p* = 0.0017), as well as the Zn 2.5 group (*p* = 0.009), whereas no significant differences in mean thickness of biofilms were detected between rhizobia strains originating from a non-polluted reference area and a metal-polluted waste heap area exposed to 1.0 mM Cd, 0.5 mM as well as 0.5 mM and 2.5 mM Pb (Table 2).

In rhizobia originating from the non-polluted Bolestraszyce grassland and the Bolesław waste heap area exposed to Cd, Zn, and Pb, significantly lower biofilm thicknesses were found in comparison to the non-exposed ones. Moreover, a dose–response effect was observed for this parameter in cases of Cd and Pb exposure; thickness of biofilms formed by rhizobia of reference and waste heap origin exposed to the highest dose of metals, i.e., 1.0 mM Cd and 2.5 mM Pb, revealed substantially thinner biofilms to those formed by rhizobia exposed to the lower dose of metals, i.e., 0.05 mM and 0.5 mM Pb (Table 2).

Strain 4.4K showed the thickest biofilms among all studied rhizobia. In comparison to other bacteria originating from the non-polluted reference Bolestraszyce grassland, it formed significantly thicker biofilms only when exposed to both concentrations of Cd. Among rhizobia strains originating from the metal-polluted Bolesław waste heap, strain 5.2H formed the thickest biofilm under non-exposed conditions. Among rhizobia of waste heap origin, strain 5.2H formed the thickest biofilm under 0.5 mM Zn exposure, while strains 7.2H and 4.1H formed the thickest biofilms when exposed to 0.5 mM Pb 0.5. No significant differences were detected in biofilm thickness under exposure to the highest doses the metals, i.e., 1.0 mM Cd, 2.5 mM Zn, and 2.5 mM Pb.

Testing of the viability of cells using the LIVE/DEAD^®^ BacLight^TM^ Bacterial Viability staining kit revealed differences in the survival of *R. leguminosarum* bv. *trifolii* cells growing in non-exposed as well as metal-exposed monospecies microsocieties. The proportion of living cells to dead cells (L/D ratio), stained with SYTO™ 9 and priopidium iodide, respectively ranged from 0.23 up to 5.58 (Table 3). No significant differences in the L/D ratios were detected between non-exposed *R. leguminosarum* bv. *trifolii* strains originating from the non-polluted reference area ($\bar{x}$_K_) and the waste heap one ($\bar{x}$_H_). However, significant differences were observed in the L/D ratios between strains of the non-polluted reference and waste heap origin when they were exposed to toxic concentrations of metals, suggesting a better survival of waste heap origin rhizobia strains when exposed to metals. Strains originating from the non-polluted reference Bolestraszyce area showed substantially lower L/D ratios as well as mean numbers of living cells (Appendix A
Appendix A) when exposed to Cd, Zn, and Pb in comparison to non-exposed cells. Moreover, a dose–response relationship was found; L/D ratios were lower when exposed to 1 mM Cd and 2.5 mM Pb in comparison to 0.05 mM Cd and 0.5 mM Pb. In biofilms of rhizobia originating from the Bolesław waste heap, the L/D ratio values were similar to those from the non-polluted reference area as well as groups where lower doses of metals were used (i.e., 0.05 mM Cd, 0.5 mM Zn, and 0.05 mM Pb). However, in biofilms of rhizobia exposed to higher doses of metals (i.e., 1.0 mM Cd, 2.5 mM Zn, and 2.5 mM Pb), substantially lower L/D ratios were observed (Table 3, Appendix A
Appendix A). The biofilm structures, with alive green-stained cells and dead red-stained ones produced by *R. leguminosarum* bv. *trifolii* strains originating from the metal-polluted site (strain 5.2H) and the non-polluted reference Bolestraszyce grassland (strain 8.8K), are presented in Figure 2.

## 3. Discussion

Analysis of the carbohydrate fractions of EPS synthesized by *R. leguminosarum* bv. *trifolii* strains originating from the about 100-year-old Zn–Pb–Cd-polluted waste heap in Bolesław as well as the non-polluted reference Bolestraszyce grassland revealed substantial differences. Among the analyzed EPS produced by the rhizobial strains growing under non-exposed conditions as well as exposed to different doses of metals, i.e., 0.05 mM and 1.0 mM Cd, 0.5 mM and 2.5 mM Zn, and 0.5 mM and 2.5 mM Pb, 30 different carbohydrates were identified, including monosaccharides, acids, alcohols, and an amino-carbohydrate (Table 1). Members of these carbohydrate groups were not found as conservative monomers of the EPS exuded by particular bacterial strains, suggesting a heteropolysaccharide type of the EPS in *R. leguminosarum* bv. *trifolii*. Skorupska et al. [54] reported that *R. leguminosarum* produces an acidic EPS, which is formed by polymerization of octa-saccharide repeating units containing glucose, glucuronic acid, and galactose in a 5:2:1 ratio. Monosaccharides and their derivatives, involving hexoses—namely, D-fructose, D-glucose, D-galactose, D-mannose, L-rhamnose, fucopyranose, amino sugars (e.g., D-glucosamine, D-galactosamine), and uronic acids (e.g., D-glucunonic acid, D-galacturonic acid, D-mannuronic acid, D-glucopyranouronic acid, D-gulopyranouronic acid, β-D-mannopyranouronic acid)—are common constituents of bacterial EPS; pentoses like D-ribose, D- and L-arabinose, and D- and L-xylose are quite rare constituents of bacterial EPS. Among heptoses, only three members of this group were found in EPS of bacteria, i.e., L-glycero*-D-*manno-heptose, D-glycero*-D-*manno-heptose, and D-glycero*-D-*galacto-heptose [55,56]. According to above, an arabinose derivative (α-D-arabinopyranose) as a single pentose was found in the EPS samples under investigation with 9.76% frequency; sedoheptulose as a new EPS unit was detected with f = 24.39%, β-talopyranose (f = 32.93), α-D,L-lyxopyranose (f = 1.22), levoglucosan (f = 10.98), and most frequently glucitol (f = 98.78) (Appendix A
Appendix A). In other organisms, for example in the fungus *Cordyceps cicadae*, EPS are composed of xylose, glucose, rhamnose, mannose, arabinose, galacturonic acid, and galactose [57], whereas alcoholic fermentation yeasts produce polysaccharides of mannans composition [58].

The observed differences in composition of polymers produced by rhizobia of waste heap origin and reference area origin (Table 1) are in concordance with other data reporting a high structural heterogeneity of synthesized EPS [54,59,60] and can confirm the hypothesis of different adaptive mechanisms that may be displayed in diverse niches and changing environmental conditions [50,52,59,61]. Our gas chromatography mass-spectrometry analyses revealed that EPS produced by *R.*
*leguminosarum* bv. *trifolii* strains isolated from nodules of *T. repens* growing on the Bolesław waste heap area and not exposed to metals evidently differ in quality of monosaccharide units in comparison to the polymers synthesized by rhizobia strains obtained from white clover growing on the non-polluted Bolestraszyce reference area. α-D,L-lyxopyranose and galactonic acid were found exclusively in EPS of rhizobia originating from the metal-polluted area, while levoglucosan, α-D-mannopyranose, and gluconic acid were characteristic for *R*. *leguminosarum* bv. *trifolii* originating from the non-polluted reference area (Table 1, Figure 1). Above results suggest that rhizobial strains of the waste heap origin possibly have realized morphological adaptations to the specific and extreme conditions of the Bolesław Zn–Pb–Cd-polluted waste heap.

The composition and quantity of EPS vary depending on the taxon, age of biofilm, and environmental conditions [62]. Availability of nutrients, extent of desiccation, pH, and oxygen fluctuations are crucial for the survival of bacteria and act forcefully under metal stress [63,64]. Our results illustrate the substantial differences in quality and quantity of carbohydrate composition of EPS metabolized extracellularly by *R. leguminosarum* bv. *trifolii* between strains of both waste heap and reference origin under metal stress. When exposed to the highest dose of Cd (1 mM), only the waste heap strains were able to produce β-talopyranose; sedoheptulose was synthesized in both groups, but significantly higher amounts were detected in EPS of strains of waste heap origin. The concentration of glucitol was significantly higher in EPS of rhizobia exposed to Cd, Zn, and Pb compared to non-exposed bacteria (Appendix A
Appendix A). Moreover, the results revealed different responses of the strains to Cd, Zn, and Pb exposure. Indeed, when exposed to the highest doses of metals, the numbers of different carbohydrates in the EPS decreased to 13% for Zn, 23–27% for Pb, and 23–30% of the total number of carbohydrates that we could identify in the EPS of rhizobia. In comparison to non-exposed, the EPS of Cd 1.0 mM exposed rhizobia did not contain carbohydrates such as levoglucosan, 6-deoxy-D-glucose, galactofuranoside, α-glucofuranose, D-galactose, α-D-glucopyranose, D-glucose, β-D-glucopyranose, and galactonic and gluconic acid; however, α-D-arabinopyranose and β-talopyranose were detected in Cd-exposed rhizobia. Moreover, EPS of waste heap origin rhizobia exposed to 0.1 mM Cd contained β-talopyranose, while EPS of unpolluted reference origin ones contained α-D-arabinopyranose, D-fructose, and β-D-galactopyranoside. In bacteria exposed to the higher dose of zinc, α-D-glucopyranoside was found as a specific EPS component to rhizobia originating from the non-polluted reference area, and sedoheptulose was found as characteristic to EPS of waste-heap origin rhizobia. Under 2.5 mM Pb exposure, rhizobial EPS were dominated by glucitol; sedoheptulose was detected in EPS of reference origin rhizobia, while β-talopyranose was specific to rhizobia originating from the Bolesław waste heap. In Gram-negative bacteria, like rhizobia, some polysaccharides are polyanionic, and the presence of uronic acids and ketal-linked pyruvates increases its anionic properties, leading to association of divalent cations [27,62]. As a consequence, heavy metals can be readily bound into EPS, which due to their involvement in flocculation can serve as a matrix of biofilms useful in bioremediation. Metal bioremediation usually is achieved by immobilization, concentration, and partitioning to an environmental compartment [65]. Nowadays, the interest to use bacteria in biofilm reactors for lowering toxic metal pollution is growing. For example, a bioremediation process has been developed using sulfate reducing bacteria (SRB) for precipitation of heavy metals from wastewater, such as *Enterobacter* sp., *Pseudomonas* sp., or *Citrobacter* sp. [16,66,67,68,69]. Considering the number of synthesized monosaccharides and their responses towards the multi-type toxic metal stress (Appendix A
Appendix A), strains 4.4K and 4.1H represent promising adapted organisms forming biofilm communities for potential application in bioremediation processes.

A multi-cellular organization as a bacterial lifestyle provides enhanced fitness for unicellular organisms in a stressful environment, especially when exposed to toxic concentrations of metals [70,71]. Analysis of the biofilms’ structure revealed significant differences in thickness of slime products of *R. leguminosarum* bv. *trifolii* strains originating from the Bolesław waste heap and the non-polluted reference area in Bolestraszyce (Table 2). Moreover, the survivability of rhizobia in these multicellular exudes substantially varies; in biofilms formed by strains originating from the waste heap, substantially higher ratios of living (green-marked)-to-dead (red-marked) cells were observed (Table 3, Appendix A
Appendix A). Biofilm and planktonic cells possess a different susceptibility towards metals [15,72,73,74], and it is suggested that chelation of metals in EPS and diminishing their transfer through the biofilm protects the “inhabitants” of a multicellular association [72]. Nonetheless, Nadell et al. [75] reported a significant role for cell–cell interactions in the formation of a biofilm structure, claiming that microevolutionary changes may have developed as a consequence of cooperative and competitive inter-cellular interactions that define biofilm form and function. Nadel et al. [75] mentioned that, when a microsociety is composed by clonemates, natural selection favors the secretion of compounds beneficial to all cells composing such microbial collective, but when different strains or species reside in a biofilm, antagonistic behavior is often favored, leading to evolution. Regarding to the characteristics of the biofilms produced by particular strains of rhizobia originating from the unpolluted reference site, strain 4.4K demonstrated significantly higher parameters of viability in harsh conditions of metal exposure in comparison to other members of this group. *R. leguminosarum* bv. *trifolii* strains originating from the Bolesław waste heap area showed a significantly higher L/D ratio than those originating from the unpolluted reference area; strain 5.2H possesses the best fitness when exposed to both doses of Cd and the low dose of zinc, while rhizobia strains 4.1H and 7.2H are doing better in cases of exposure to the higher dose of Pb ions (Table 3, Appendix A
Appendix A).

Altogether, our results indicate significant differences in EPS carbohydrate profiles and strongly confirm the presence of morphological adaptations in EPS layers resulting in biofilm characteristics, which have a role in coping with the toxic metals concentrations present in the waste heap area of Bolesław. Taking into consideration that the synthesis of EPS mostly is under the control of genes [52,76], we can postulate a genetic basis for the metal tolerance in the rhizobia strains from waste heap origin.

## 4. Materials and Methods

In total, seven strains of rhizobia isolated from root nodules of white clover (*T. repens*) growing on the about 100-year-old old Bolesław waste heap (Silesia-Krakow Upland, S. Poland, 50°17′ N 19°29′ E) and the non-polluted reference Bolestraszyce grassland area (Bolestraszyce Foothills, S-E Poland, 49°48 ′N 22°50 ′E), representing different genotypes of *R*. *leguminosarum* bv. *trifolii* [38], were used in the study. *Rhizobium* strains were explored for quantitative and qualitative characteristics of the carbohydrate fraction of exopolysaccharides (EPS) using gas chromatography mass-spectrometry method (GC-MS), as well as for biofilms formation (colorimetric crystal violet method) and structure analysis using confocal laser scanning microscopy (CLSM).

Three strains of *R*. *leguminosarum* bv. *trifolii* isolated from root nodules of *T. repens* growing on the metal-polluted site, i.e., 4.1H, 5.2H, 7.2H, and four strains from the reference origin, i.e., 3.9K, 4.4K, 5.10K, 8.8K, were investigated without (non-exposed control group) and with exposure to different Zn, Pb, and Cd concentrations. The strains were incubated in liquid 79CA medium (10 g L^−1^ D-mannitol, 0.5 g L^−1^ KH_2_PO_4_, 0.2 g L^−1^ MgSO_4_ × 7H_2_O, 0.1 g L^−1^ NaCl, 0.1 g L^−1^ calcium glycerophosphate, 1 g L^−1^ yeast extract Difco, 1 g L^−1^ casein acid hydrolysate, pH 7.2–7.4) [77], which was supplemented with metals at concentrations established based on literature data [78,79] as follows: 0.5 mM (Zn 0.5 group) and 2.5 mM ZnSO_4_ × 7H_2_O (Zn 2.5 group), 0.5 mM (Pb 0.5 group) and 2.5 mM Pb(NO_3_)_2_ (Pb 2.5 group), 0.05 mM (Cd 0.05 group) and 1.0 mM CdCl_2_ × 2.5H_2_O (Cd 1.0 group).

### 4.1. Exopolysaccharide (EPS) Carbohydrates Analysis

#### 4.1.1. EPS Extraction

In order to isolate EPS, the *R. leguminosarum* bv. *trifolii* strains were cultured in liquid 79CA medium supplemented or not (non-exposed control group) with two doses of Cd, Zn, and Pb for four days on a rotary shaker at 160 rpm and 28 °C. After removal of the rhizobia by centrifugation at 8000× *g* for 20 min, 3 volumes of cold pure 99.9% absolute ethanol (EtOH, Emplura^®^, Sigma Aldrich, St. Louis, MO, USA) were added to the supernatant and incubated for 24 h at 4 °C. The EPS precipitate was centrifuged at 4000× *g* for 20 min. The pellet was washed three times with 0.5 mL ultrapure (MilliQ) water and dried in a centrifugal vacuum concentrator (Eppendorf^®^, Hamburg, Germany) at 13,000× *g* for 12 h.

#### 4.1.2. EPS Monosaccharides Analysis by GC-MS

Evaluation of the EPS carbohydrate fraction was performed as qualitative and quantitative analysis of trimethylsilyl-ester O-methyl glycolsyl—derivatives, which arise as products of acidic methanolysis and derivatization (silylation) of polysaccharides [79,80].

Each sample was put into quartz vials and methanolyzed with 1.5 mL anhydrous 1M methanol containing 0.5M HCl. Hydrolysis of monosaccharides to their corresponding methyl glycosides was performed at 80 °C ± 5 for 18 h on a sand bath. The excess of reagent was evaporated under nitrogen gas flow for 4 h at room temperature. Then, the methyl glycosides were converted to their trimethylsilyl derivatives. This was performed by adding 340 µL anhydrous pyridine (99.8%, Sigma Aldrich, St. Louis, MO, USA) and 80 µL BSTFA (N,O-bis(trimethylsilyl)trifluoroacetamide, Sigma Aldrich, St. Louis, MO, USA) to each dry sample [57,58]. The mixture was incubated for 30 min on a sand bath at 80 °C, and excess of the reagent was removed under a stream of nitrogen gas at room temperature. In parallel, different carbohydrate standards (D-(-)fructose, L-rhamnose monohydrate, D-(+)-galactose, D-glucuronic acid, D-(+)-glucose, *myo*-inositol, D-(+)-galacturonic acid monohydrate, D-(+)-xylose; Sigma Aldrich, St. Louis, MO, USA) were converted to their corresponding derivatives for analysis and obtaining patterns crucial to identification and calibration of the standard curves. Samples were dissolved in 500 µL of *n*-hexane (Sigma Aldrich MS SupraSolv^®^, St. Louis, MO, USA). One microliter of each sample was analyzed in an Agilent Technologies GC 6890N instrument (Santa Clara, CA, USA) with HP Mass Selective Detector (MSD 5973, Santa Clara, CA, USA), equipped with a fused silica capillary column (30 m × 0.25 mm ID) with a 5% phenyl methylsiloxane (HP-5MS) of 0.25 µm thickness stationary phase. An autosampler (HP 7683) and a split/splitless injector (injector temperature was 250 °C) were used. Helium of 99.999% purity was used as a carrier gas at a flow rate 1 mL min^−1^ and a split ratio 1:20. The oven temperature program started from 120 °C and increased with 1 °C min^−1^ up to 180 °C and thereafter 10 °C min^−1^ up to 300 °C. The total run time was 72 min with the 7 min solvent delay time. The electron ionization (EI) mass spectrometer worked in scan mode from 41 to 650 amu (atomic mass unit) with electron energy 70 eV. The electron source temperature equaled 230 °C, the quadrupole and GC interface temperatures were 150 °C and 280 °C, respectively.

##### Qualitative Assessment of EPS Monosaccharides

Identification of *R*. *leguminosarum* bv. *trifolii* EPS monosaccharides was performed on the basis of the mass spectrum and retention parameters registered during GC-MS measurement. Determined retention indices of studied samples were compared with retention indices of studied standards and mass spectra. When retention indices and mass spectra of monosaccharides obtained from rhizobial EPS were not similar to those registered for standards, the values compiled in the relevant databases were used for comparison; the NIST 11 MS spectra database, which comprises the Agilent Technologies GC System HP 6890N, NIST Chemistry WebBook [81] (retention indices), and Isidorov [82], was used. Retention indices of the monosaccharides were calculated using retention times determined during analysis of C_12_-C_33_
*n*-alkanes in identical conditions. The calculations were performed according to the Van den Dool and Kratz [83] Equation (1) for arithmetic retention index (*I*). Two subsequent *n*-alkanes were chosen to fulfill the following condition t_n_≤ t_x_ ≤ t_n+1_.
*I* = 100 [(t_x_ − t_n_)/(t_n+1_ − t_n_) + n](1)
where:

t–retention time of an analyzed compound

t_n_–retention time of *n*-alkane leaving the chromatographic column before the analyzed compound

t_n+1_–retention time of *n*-alkane leaving the chromatographic column after the analyzed compound

###### Quantitative Assessment of EPS Monosaccharides

Monosaccharide concentrations in the samples were estimated based on the calibration curves performed using eight saccharide standards, i.a., D-(-)fructose, L-rhamnose monohydrate, D-(+)-galactose, D-glucuronic acid, D-(+)-glucose, myo-inositol, D-(+)-galacturonic acid monohydrate, D-(+)-xylose (Sigma Aldrich) (Appendix A
Appendix A). In order to prepare standard curves, the solutions were made as a mixture of particular standard monosaccharides in concentrations of 10 µg mL^−1^, 50 µg mL^−1^, 100 µg mL^−1^, 200 µg mL^−1^, and 500 µg mL^−1^. Each solution was analyzed in three replicates. The method validation parameters were determined, i.e., variation coefficient (*CV*), limit of determination (*LOD*) and limit of quantification (*LOQ*), and calculated using the Equations (2)–(4) as follows:(2)$$CV\;=\frac{SD}{\bar{x}}\cdot 100\%$$
where:

SD–standard deviation

$\bar{x}$–arithmetic mean
(3)$$LOD=\frac{3\;\times \;SD}{a}$$
where:

a–slope of standard curve equation
(4)$$LOQ\;=\frac{10\;\times \;SD}{a}$$

In order to distinguish highly similar monosaccharides without appropriate standards, the calibration curves of standards with the highest identity to the chemical structure were used. Therefore, to calculate the glucitol content a *myo*-inositol standard curve was used, for glucuronic acid a D-glucuronic acid, for α-D-glucopyranoside a D-(+)-glucose, and for β-D-galactopyranose a D-(+)-galactose; for D-rhamnose a L-rhamnose monohydrate calibration curve was used.

### 4.2. Biofilm Formation and Structure Analysis

The capability of the *R*. *leguminosarum* bv. *trifolii* strains to form biofilms was examined colorimetrically using the crystal violet method [53,84,85,86,87], and the biofilm structure was estimated with using confocal laser scanning microscopy (CLSM) [76,88].

#### 4.2.1. Quantification of Biofilm Formation

The absorbance of three *R*. *leguminosarum* bv. *trifolii* strains from the Bolesław waste heap and the four from the non-polluted reference Bolestraszyce area after 24 h cultivation at 28 °C in 79CA liquid medium supplemented or not with different doses of metals was as follows: 0.5 mM (Zn 0.5 group) and 2.5 mM (Zn 2.5 group), 0.5 mM (Pb 0.5 group) and 2.5 mM (Pb 2.5 group), 0.05 mM (Cd 0.05 group) and 1.0 mM (Cd 1.0 group) was adjusted with the appropriate 79CA medium mixture up to 0.6 optical density (OD) units using a spectrophotometer at wavelength λ = 560 nm (OD_560_ = 0.6), and diluted 100-fold. The 100 µL aliquots were transferred in six repetitions into 96-well sterile microtitration cell culture plates (NEST Scientific, Rahway, NJ, USA) and incubated for 3 days at 28 °C in static conditions. After incubation, the medium was aspired and discarded, and the deposit that consisted of aggregated rhizobia cells (biofilm) was rinsed tree times with 100 µL sterile 1 × PBS buffer (137 mM NaCl, 2.7 mM KCl, 10 mM Na_2_HPO_4_, 1.8 mM KH_2_PO_4_ in 1.0 L, pH 7.4). Biofilms were treated with 100 µL of 0.4% crystal violet water solution for 30 min. After staining, the dye was discarded, and the biofilm sediments in each well were rinsed three times with sterile distilled water, air-dried at room temperature, and flooded with a mixture of 95% ethanol and acetone (7:3, *v*/*v*). Biofilm formation was considered as positive at OD_490_ > 0.250 with SpectraMax M2 Series Multi-Mode Microplate Reader (SoftMax Pro, Molecular Devices LLC, San Jose, CA, USA) [89].

#### 4.2.2. Biofilms Structure Analysis

The *R. leguminosarum* bv. *trifolii* strains originating from the Bolesław metal-polluted waste heap and the Bolestraszyce non-polluted reference area were cultured in 79CA liquid medium appropriately supplemented or not with two different doses of Cd, Zn, and Pb (see above) in a rotary shaker (160 rpm) for 24h at 28 °C. Next, the absorbance of the cultures was adjusted to 0.6 at OD_600_ and diluted 100-fold in the same sterile 79CA medium with or without Cd, Zn, and Pb. A 100 µL aliquot of each bacterial sample was transferred into three wells of 96-well polystyrene sterile microwell plates (NEST Scientific) and incubated for 4 days at 28 °C under static conditions. After that, culture liquid media were discarded from each well of the plate, and sedimented bacteria were washed gently three times with sterile 200 µL of 0.85% NaCl. Subsequently, the *Rhizobium* biofilms were stained in darkness for 30 min at room temperature with 1.5 µL of the mixture composed of equal volumes of two components LIVE/DEAD^®^ BacLight™ Bacterial Viability stain kit (Invitrogen, Thermo Fisher Scientific, Waltham, MA, USA): 1.67 mM Syto^TM^ 9 and 1.67 mM propidium iodide (PI) (component 1) and 1.67 mM Syto™ 9 and 18.3 mM propidium iodide (component 2) in 0.85% NaCl, according to the manufacturer’s instructions. SYTO^®^ 9 labels bacteria both with intact and damaged membranes, giving their nucleic acids green fluorescence with excitation/emission maxima at 480/500 nm wavelength, but when SYTO^®^ 9 is used together with propidium iodide, the latter dye enters only bacterial cells with damaged membranes, causes a SYTO^®^ 9 fluorescence reduction in cells with damaged membranes, and stains their nucleic acids with red fluorescence at wavelength maxima 490/635 nm. After removal of dyes not bound to cells, the formed biofilms were rinsed three times with sterile 0.85% NaCl and analyzed with a Confocal Laser Scanning Microscope LSM 5 PASCAL (Carl Zeiss, Jena, Germany). The images of biofilms were recorded using an inverted fluorescence microscope AxioVert 200M (Carl Zeiss, Jena, Germany) with a scanning head LSM 5 PASCAL (Carl Zeiss, Jena, Germany). The biofilm thickness was recorded using AIM 4.2 software (Carl Zeiss, Jena, Germany) in a multifaceted laser scan mode. The images were recorded using the AxioVision 4.8 (Carl Zeiss, Jena, Germany) software, and the ratio of living/dead cells was determined by the multichannel fluorescence technique using an AxioCam HR3 camera and 470 nm and 546 nm filters for the green and red channels, respectively. The ratio of live to dead cells was calculated using the ImageJ 1.50i software (Wayne Rasband, NIH, USA) [90,91]. In order to determine the percentages of living and dead cells (L/D ratio) in biofilms, ten independent sets of images were collected from three wells for each strain.

### 4.3. Statistical Analysis

The results are presented as means ± SD (standard deviation), analyzed with one-way analysis of variance (ANOVA), and significant differences between means were estimated using the multiple range Duncan’s test using Statistica ver. 13.3 TIBCO (Palo Alto, CA, USA).

## 5. Conclusions

The results obtained in the study clearly show the substantial differences in EPS carbohydrates composition as well as the differences in survival of bacterial “inhabitants” in biofilms produced by *R. leguminosarum* bv. *trifolii* strains originating from a non-polluted reference area and a metal-polluted waste heap site. Significant differences in EPS composition and biofilm structure were also detected among these groups of strains when exposed to Zn, Pb, and Cd, suggesting the presence of adaptations in EPS resulting in biofilm formation. Identification of EPS and biofilm adaptations in the Zn–Pb waste heap inhabiting *R. leguminosarum* bv. *trifolii* strains enhance our understanding of the basic mechanisms of the rhizobial metal-tolerance system and is promising for application in bioremediation.

## Figures and Tables

**Figure 1 ijms-22-02808-f001:**
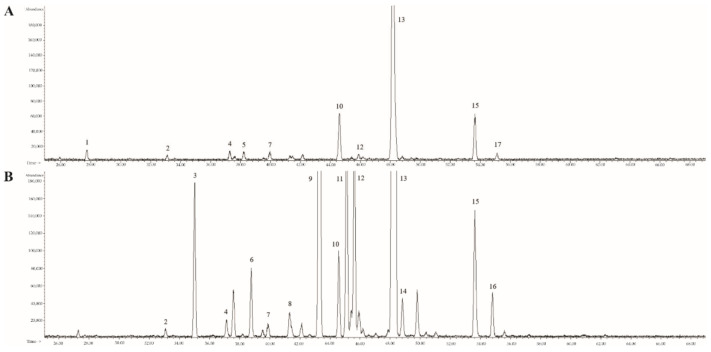
Chromatograms of carbohydrates received by methanolysis of exopolysaccharides isolated from *R. leguminosarum* bv. *trifolii* deriving from nodules of white clover growing on a non-polluted reference (**A**) and Zn–Pb waste heap area (**B**). Abbreviations: 1, levoglucosan; 2, β-D-xylopyranose; 3, 6-deoxy-D-glucose; 4, galactofuranoside; 5, α-D-mannopyranose; 6, D-fructose; 7, β-D-galactopyranoside; 8, α-glucofuranose; 9, D-galactose; 10, α-D-glucopyranoside; 11, α-D-glucopyranose; 12, D-glucose; 13, glucitol; 14, sedoheptulose; 15, β-D-glucopyranose; 16, galactonic acid; 17, gluconic acid.

**Figure 2 ijms-22-02808-f002:**
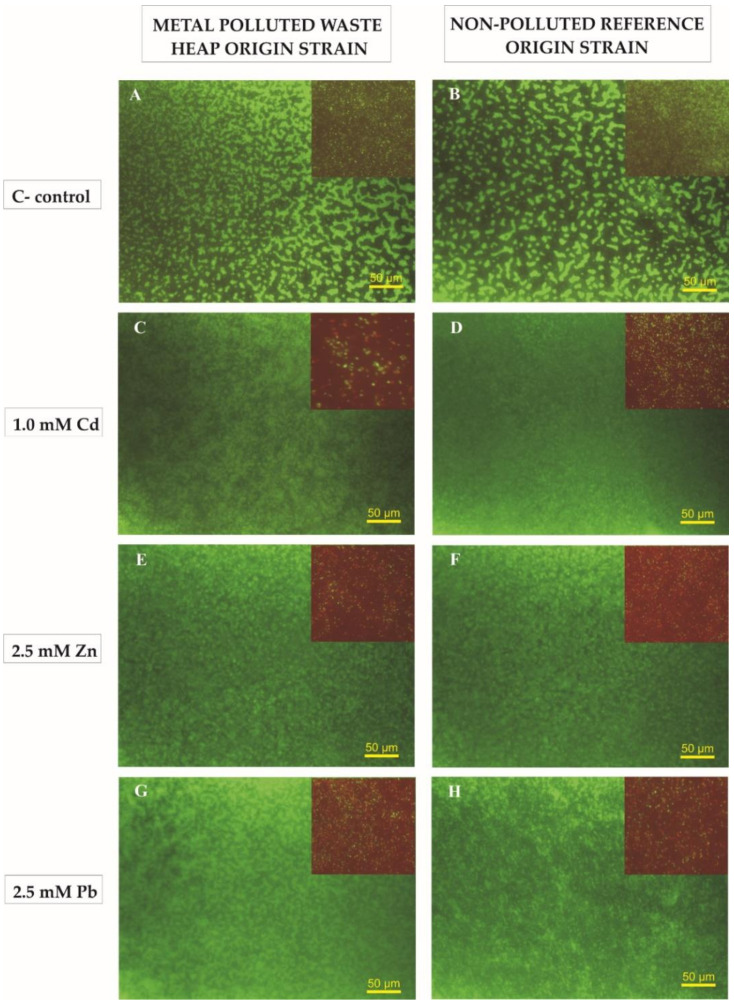
Biofilms structure of *R. leguminosarum* bv. *trifolii* strains originating from the metal-polluted Zn–Pb Bolesław waste heap (5.2 H) and non-polluted reference Bolestraszyce area (8.8 K) under control conditions (**A**,**B**) as well as under 1.0 mM Cd (**C**,**D**), 2.5 mM Zn (**E**,**F**), and 2.5 mM Pb (**G**,**H**) stress-stained with Syto^TM^ 9 (a green fluorescence) and propidium iodide (a red fluorescence). Scale bar corresponds to the length of 50 µm.

**Table 1 ijms-22-02808-t001:** Qualitative and quantitative (µg µL^−1^) carbohydrate composition of exopolysaccharide *Rhizobium leguminosarum* bv. *trifolii* strains isolated from root nodules of *Trifolium repens* growing on the non-polluted reference grassland in Bolestraszyce (R) and the metal-polluted waste heap in Bolesław (WH) not exposed (control) and exposed to 0.05 mM Cd (Cd 0.05) and 1.0 mM (Cd 1.0), 0.5 mM Zn (Zn 0.5) and 2.5 mM Zn (Zn 2.5), and 0.5 mM Pb (Pb 0.5) and 2.5 mM Pb (Pb 2.5).

Carbohydrate	Control		Cd 0.05		Cd 1.0		Zn 0.5		Zn 2.5		Pb 0.5		Pb 2.5	
	R	WH	R	WH	R	WH	R	WH	R	WH	R	WH	R	WH
α-D,L-lyxopyranose	-	15.13 ^a^	-	-	-	-	-	-	-	-	-	-	-	-
α-D-arabinopyranose	-	-	-	-	17.78 ^a^	-	21.70 ± 1.60 ^a^	-	-	-	51.50 ± 43.56 ^a^	18.78 ± 1.19 ^a^	-	-
D-rhamnose	-	-	-	-	-	-	-	-	-	-	-	6.69 ^b^	-	-
L-rhamnose monohydrate	-	-	-	-	-	-	6.41 ^b^	-	-	-	12.02 ^a^	-	-	-
levoglucosan	18.84 ± 0.22 ^a^	-	18.90 ± 1.05 ^a^	17.92 ± 0.44 ^a^	-	-	-	-	-	-	-	-	-	-
6-deoxy-D-glucose	55.00 ± 1.33 ^b^	40.47 ± 21.27 ^a^	-	-	-	-	-	-	-	-	-	-	-	-
methyl α-D-mannopyranoside	19.72 ± 0.22 ^a^	20.00 ± 0.64 ^a^	-	-	47.09 ± 13.79 ^b^#	32.68 ± 3.24 ^a^#	31.36 ± 8.66 ^a^	36.61 ± 12.86 ^a^	25.29 ± 8.78 ^a^	18.21 ± 0.25 ^a^	76.63 ± 11.57 ^b^#	73.08 ± 27.78 ^c^#	47.42 ± 7.41 ^a^#	63.22 ± 17.38 ^a^#
galactofuranoside	20.88 ± 1.24 ^a^	18.72 ± 0.11 ^a^	15.07 ± 3.48 ^a^	-	-	-	18.65 ± 0.17 ^a^	-	-	-	18.43 ^a^	-	-	-
α-D-mannopyranose	19.41 ± 0.15 ^a^	-	20.73 ± 2.02 ^a^	18.55 ^a^	18.66 ± 0.49 ^a^	18.15 ± 0.13 ^b^	19.08 ± 0.74 ^a^	19.53 ± 1.56 ^b^	-	-	21.77 ± 0.82 ^a^	21.55 ± 3.75 ^a^	18.86 ± 0.16 ^b^	19.35 ± 0.73 ^b^
D-fructose	32.72 ± 2.55 ^c^	40.13 ± 9.54 ^b^	-	-	15.10 ± 2.45 ^a^	-	13.78 ± 1.56 ^c^#	21.00 ± 6.66 ^b^#	13.58 ^a^#	11.88 ± 0.27 ^b^#	16.73 ± 3.51 ^a^#	20.38 ± 13.77 ^a^#	12.89 ± 0.80 ^b^#	13.56 ± 2.13 ^b^#
β-D-galactopyranoside	19.43 ± 0.28 ^a^	16.81 ± 2.22 ^a^	13.75 ± 1.20 ^a^	19.79 ^a^	23.55 ^a^	-	87.08 ± 59.59 ^a^#	18.54 ± 0.85 ^b^*	-	-	69.51 ± 64.05 ^a^#	36.57 ± 23.88 ^ab^#	-	18.62 ^b^
α-D-galactofuranose	-	-	-	-	-	-	18.80 ^a^	-	-	-	19.21 ^a^	-	-	-
α-D-mannofuranose	-	-	-	-	-	-	14.74 ± 2.12 ^c^	-	-	-	-	13.65 ^a^	-	-
α-glucofuranose	20.11 ± 2.28 ^a^	23.52 ± 1.15 ^a^	18.06 ^a^	-	-	-	-	18.33 ± 0.75 ^b^	-	-	-	18.86 ^a^	-	-
D-galactose	235 ± 8.42 ^d^	172 ± 144 ^b^	18.52 ^a^#	-	-	-	25.11 ± 4.78 ^a^#	-	-	-	23.65 ± 7.38 ^a^#	22.73 ± 7.07 ^a^#	-	-
α-D-glucopyranoside	23.09 ± 3.88 ^a^	21.89 ± 5.40 ^a^	24.64 ± 3.77 ^a^	23.78 ± 5.20 ^a^	20.84 ± 2.52 ^a^	21.27 ± 2.64 ^b^	40.15 ± 24.22 ^a^	39.19 ± 15.94 ^a^	19.95 ± 0.03 ^a^	-	84.63 ± 31.64 ^a^#	123.96 ± 131 ^abc^#	20.38 ± 0.73 ^b^	22.63 ± 3.24 ^b^
α-D-glucopyranose	66.42 ± 14.6 ^b^	75.29 ± 15.54 ^c^	-	-	-	-	-	-	-	-	-	-	-	-
D-glucose	58.33 ± 30.1 ^b^	69.23 ± 16.04 ^c^	18.12 ^a^#	18.16 ^a^#	-	-	21.26 ± 3.49 ^a^#	19.20 ± 1.61 ^b^#	-	-	19.70 ± 0.82 ^a^	54.73 ± 6.74 ^c^*	-	-
β-D-galactopyranose	18.88 ± 3.15 ^a^	16.64 ± 0.00 ^a^	-	-	19.00 ± 1.04 ^a^	18.76 ± 0.77 ^b^	25.14 ± 7.54 ^a^	23.72 ± 5.21 ^b^	-	-	42.86 ± 12.49 ^a^#	60.80 ± 53.75 ^abc^	18.48 ± 0.16 ^b^	19.19 ± 1.03 ^b^
β-D-mannopyranose	-	-	-	-	-	-	18.32 ± 0.35 ^a^	19.48 ± 1.38 ^b^	-	-	19.56 ± 0.46 ^a^	24.57 ± 7.45 ^ab^	-	-
β-talopyranose	-	-	-	-	-	20.54 ^b^	41.17 ± 21.08 ^a^	47.71 ± 15.27 ^a^	-	-	174.68 ± 55.58 ^b^	56.45 ± 22.51 ^ac^	-	18.69 ^b^
sedoheptulose	20.26 ± 3.40 ^a^	24.12 ± 3.58 ^a^	-	-	134 ± 106 ^b^#	187 ± 2.57 ^c^#	-	11.18 ^b^	-	17.43 ^a^	21.30 ^a^	-	28.58 ^ab^	-
β-D-glucopyranose	24.90 ± 4.98 ^a^	24.23 ± 9.10 ^a^	25.82 ± 2.86 ^a^	24.64 ± 4.58 ^a^	-	-	-	-	-	-	-	-	-	-
gluconic acid	47.47 ± 2.99 ^b^	-	53.40 ± 3.15 ^b^	48.37 ^b^	-	-	71.37 ^d^	50.34 ± 4.33 ^a^	-	-	46.54 ^a^	-	-	-
glucuronic acid	-	-	-	-	-	-	64.31 ± 31.74 ^a^	70.10 ± 25.83 ^a^	-	-	92.74 ± 36.33 ^b^	65.60 ± 27.40 ^bc^	-	-
D-glucuronic acid	-	-	-	-	-	-	41.91 ^a^	-	-	-	-	-	-	-
*myo-inositol*	-	-	-	-	-	-	-	-	-	-	26.93 ^a^	-	-	-
N-acetyl-D-glucosamine	-	-	-	-	-	-	17.52^a^	-	-	-	19.85 ^a^	-	-	-
glucitol	250 ± 233 ^d^	327 ± 322 ^b^	84.58 ± 35.73 ^b^#	86.37 ± 20.44 ^b^#	1145 ± 240 ^c^#	717 ± 99 ^d^*#	575.05 ± 186.7 ^c^#	761.54 ± 67.24 ^c^#	654.79 ± 93.12 ^b^#	633.23 ± 53.83 ^c^#	1562 ± 234 ^c^#	948.12 ± 451 ^d^	868.81 ± 135 ^c^#	1181 ± 157.83 ^c^#
galactonic acid	-	179 ± 175 ^b^	-	-	-	-	177.29 ± 112 ^a^	112.90 ± 76.86 ^d^	-	-	133.11 ± 34.75 ^b^	1532 ± 244 ^d^*	-	-

Results are presented as means ± SD for *n* = (2–8), analyzed with one-way analysis of variance (ANOVA), and significant differences between means were estimated with the usage of the multiple range Duncan’s test. Letters in upper case are used to mark significance of differences among rhizobia of non-polluted reference area origin and from the metal-polluted waste heap site; an asterisk is used to mark significant differences between non-polluted reference area and waste heap origin strains; within each particular group, a number sign is used to mark significance of differences between bacteria exposed to a particular metal dose vs. non-exposed control group. Not detected compounds are indicated as a hyphen. A range of colors from yellow to green is proportional to the concentration increase of particular carbohydrates, with the exception of the compounds of the highest concentrations (up to over 1000 µg µL^–1^) where red-scale intensity was applied.

**Table 2 ijms-22-02808-t002:** Thickness of biofilms (µm) formed by *R. leguminosarum* bv. *trifolii* strains isolated from root nodules of *T. repens* growing on non-polluted reference grassland Bolestraszyce (K) and 100-year-old Zn–Pb waste heap Bolesław (H) not exposed (C) and exposed to 0.05 mM Cd (Cd 0.05) and 1.0 mM (Cd 1.0), 0.5 mM Zn (Zn 0.5) and 2.5 mM Zn (Zn 2.5), and 0.5 mM Pb (Pb 0.5) and 2.5 mM Pb (Pb 2.5).

Rhizobial Strain	C	Cd 0.05	Cd 1.0	Zn 0.5	Zn 2.5	Pb 0.5	Pb 2.5
8.8K	42.66 ± 7.51 ^a^	27.88 ± 1.81 ^a^	17.28 ± 1.10 ^a^	18.40 ± 2.07 ^a^	19.33 ± 2.10 ^a^	39.53 ± 0.00 ^a^	11.86 ± 0.06 ^a^
4.4K	76.26 ± 7.90 ^b^	36.09 ± 2.70 ^b^	19.12 ± 1.08 ^b^	20.63 ± 0.98 ^a^	22.62 ± 2.97 ^a^	39.43 ± 2.09 ^a^	12.44 ± 0.01 ^a^
3.9K	61.99 ± 1.01 ^a^	33.80 ± 2.80 ^a^	16.36 ± 0.60 ^a^	19.41 ± 0.45 ^a^	21.28 ± 2.58 ^a^	38.63 ± 2.08 ^a^	12.98 ± 0.25 ^a^
5.10K	43.68 ± 6.45 ^a^	29.75 ± 3.35 ^a^	17.31 ± 1.21 ^a^	18.22 ± 3.08 ^a^	20.45 ± 2.18 ^a^	37.59 ± 1.18 ^a^	11.19 ± 0.09 ^a^
$\bar{x}$ **_K_**	**59.46 ± 17.65**	**31.99 ± 3.94 #**	**18.20 ± 1.40 ##**	**19.52 ± 1.90 #**	**20.98 ± 2.92 #**	**39.46 ± 1.71 #**	**12.09 ± 0.30 ##**
5.2H	47.82 ± 6.26 _a_	27.39 ± 6.58 _a_	16.45 ± 1.68 _a_	37.86 ± 2.87 _a_	15.07 ± 1.12 _a_	19.03 ± 1.33 _a_	10.94 ± 0.81 _a_
7.2H	27.44 ± 2.60 _b_	26.25 ± 2.30 _a_	17.22 ± 0.25 _a_	25.00 ± 4.44 _b_	17.62 ± 0.66 _a_	28.64 ± 4.60 _b_	14.96 ± 1.11 _a_
4.1H	35.54 ± 11.22 _b_	24.06 ± 2.97 _a_	17.17 ± 1.01 _a_	19.73 ± 0.14 _c_	16.27 ± 0.67 _a_	41.95 ± 15.95 _b_	14.15 ± 0.25 _a_
$\bar{x}$ **_H_**	**32.40 ± 5.04 ***	**23.62 ± 4.06 *#**	**16.30 ± 1.09 ##**	**23.00 ± 8.50 #**	**15.81 ± 1.33 *#**	**27.06 ± 12.98 #**	**12.76 ± 1.97 ##**

Results are presented as means ± SD for *n* = 30, analyzed with one-way analysis of variance (ANOVA). Significant differences between means were estimated using the multiple range Duncan’s test. Abbreviations: $\bar{x}$_K_, mean biofilm thickness in a rhizobia non-polluted reference origin group; $\bar{x}$_H_, mean biofilm thickness in a rhizobia metal-polluted waste heap origin group. Asterisk was used to mark significance of differences between non-polluted reference area and waste heap origin strains; letters in upper case were used to mark significance of differences among rhizobia of non-polluted reference area origin; letters in lower case were used to indicate significance of differences among waste heap area origin; a number sign was used to indicate the significance of differences between control and two doses of particular metal, i.e., Cd 0.05 vs. Cd 1.0, Zn 0.5 vs. Zn 2.5, and Pb 0.5 vs. Pb 2.5 group. Density of colors proportional with an increase of biofilm thickness was used to mark differences between particular rhizobial strains (green) and between non-polluted reference origin bacteria and waste heap ones (red).

**Table 3 ijms-22-02808-t003:** Living-to-dead cell ratio (L/D) in biofilms formed by *R*. *leguminosarum* bv. *trifolii* strains isolated from root nodules of *T. repens* growing on non-polluted reference grassland BolesTable 100. year-old Zn–Pb waste heap Bolesław (H) non-exposed (C) and exposed to 0.05 mM (Cd 0.05) and 1.0 mM (Cd 1.0), 0.5 mM (Zn 0.5) and 2.5 mM (Zn 2.5), and 0.5 mM (Pb 0.5) and 2.5 mM (Pb 2.5).

Rhizobial Strain	C	Cd 0.05	Cd 1.0	Zn 0.5	Zn 2.5	Pb 0.5	Pb 2.5
8.8K	3.19 ± 0.29 ^a^	2.00 ± 0.20 ^a^	0.37 ± 0.09 ^a^	0.32 ± 0.03 ^a^	0.53 ± 0.11 ^a^	2.18 ± 0.33 ^a^	0.48 ± 0.12 ^a^
4.4K	2.94 ± 0.34 ^a^	2.58 ± 0.31 ^a^	0.96 ± 0.04 ^b^	0.78 ± 0.30 ^b^	1.33 ± 0.09 ^b^	3.75 ± 0.35 ^b^	0.80 ± 0.05 ^b^
3.9K	2.52 ± 0.06 ^a^	1.45 ± 0.03 ^b^	0.23 ± 0.01 ^c^	0.38 ± 0.06 ^a^	0.28 ± 0.08 ^a^	1.63 ± 0.24 ^a^	0.36 ± 0.07 ^a^
5.10K	3.25 ± 0.16 ^a^	2.23 ± 0.12 ^a^	0.35 ± 0.08 ^a^	0.32 ± 0.08 ^a^	0.52 ± 0.09 ^a^	2.18 ± 0.21 ^a^	0.48 ± 0.11 ^a^
$\bar{x}$ **_K_**	**3.43 ± 0.51**	**2.07 ± 0.15 #**	**0.34 ± 0.10 ##**	**0.32 ± 0.07 #**	**0.36 ± 0.17 #**	**2,75 ± 0.21 #**	**0.42 ± 0.05 ##**
5.2H	2.94 ± 1.21 _a_	4.69 ± 1.18 _a_	1.59 ± 0.17 _a_	4.26 ± 0.54 _a_	0.52 ± 0.06 _a_	4.80 ± 0.84 _a_	0.56 ± 0.02 _a_
7.2H	3.01 ± 0.60 _a_	2.72 ± 0.57 _b_	1.19 ± 0.06 _b_	2.67 ± 0.44 _b_	1.34 ± 0.12 _b_	3.62 ± 0.63 _a_	0.73 ± 0.05 _b_
4.1H	3.06 ± 0.72 _a_	2.15 ± 0.07 _b_	0.43 ± 0.02 _c_	0.41 ± 0.07 _c_	0.41 ± 0.03 _c_	5.58 _a_	0.50 ± 0.01 _c_
$\bar{x}$ **_H_**	**3.00 ± 0.69**	**3.14 ± 0.87 ***	**1.21 ± 0.27 *#**	**3.10 ± 0.85 ***	**0.89 ± 0.27 *#**	**4.01 ± 0.71 ***	**0.57 ± 0.04 *#**

Results are presented as means ± SD for *n* = 30, analyzed with one-way analysis of variance (ANOVA). Significant differences between means were estimated with the usage of the multiple range Duncan’s test. Abbreviations: $\bar{x}$_K_, mean L/D ratio in a rhizobia non-polluted reference origin group; $\bar{x}$_H_, mean L/D ratio in a rhizobia metal-polluted waste heap origin group. Asterisk was used to mark significance of differences between non-polluted reference and waste heap origin strains; letters in upper case were used to mark significance of differences among rhizobia of non-polluted reference origin; letters in lower case were used to indicate significance of differences among waste heap area origin a number sign was used to indicate the significance of differences between control and two doses of particular metal, i.e., Cd 0.05 vs. Cd 1.0, Zn 0.5 vs. Zn 2.5, and Pb 0.5 vs. Pb 2.5 group. Density of colors proportional with an increase of L/D ratio values was used to mark differences between particular rhizobia strains (green) and between non-polluted reference origin bacteria and waste heap ones (red).

## Data Availability

The data presented in this study are available in the article.

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
