# Peer review of "Exopolysaccharide Carbohydrate Structure and Biofilm Formation by Rhizobium leguminosarum bv. trifolii Strains Inhabiting Nodules of Trifoliumrepens Growing on an Old Zn–Pb–Cd-Polluted Waste Heap Area"

_ijms, 2021, doi:10.3390/ijms22062808_

Round 1
Reviewer 1 Report
General comments
The manuscript by Olenska et al. represents a detailed investigation of the EPS produced by Rhizobium isolated from a heavy metal contaminated site. The authors compare the effect of different metal concentrations in relation to the EPS produced by Rhizobium from an unpolluted site. The carbohydrates from isolated EPS isolated was subject to GC-MS analysis and various preparative and derivatisation procedures. Biochemical analysis was done qualitatively and quantitatively. The methods and results are meticulously described. In addition the authors examined biofilm formation by a traditional staining technique as well as fluorescent staining in combination with confocal laser scanning microscopy.
Overall the manuscript is written / organised well and straight forward. However, there are several items to address before publication.
Detailed comments
Biofilm development in 96 well plates – this technique is used by several scientists in the field, however its always the question whether this is sedimentation and sticking of cells/aggregates to a surface OR is it real biofilm development and growth, furthermore biofilms in many cases are growing under some kind of flow and shear force – Nevertheless, in this case we may look into growth somehow as the plates were incubated for 3 days
Live/Dead stain – Similarly to the 96 well issue, the L/D staining kit is judged critically, there are believers and others saying it is nonsense – However, I assume that the authors applied it properly in order to get reliable results. Usually with pure cultures the results are more meaningful.
BUT
Its not clear at all why the 96 well grown biofilms were first dried before they were stained with L/D? The same question comes up if we look into confocal microscopy? The big advantage of this 3d technique lies in the fact that biological samples can be examined in the natural hydrated state!
This brings me to the next point – the confocal datasets are very poor in quality!
Questions:
- What are we looking at? Maximum intensity projections? There is no biofilm “architecture” visible. It might be better to project the datasets in a different way.
- Why is there such a high background? This may originate from the 30 min drying which will shrink the original 3d biofilm structure into a more condensed mass of cells.
- Why is the resolution so poor? Probably due to a low magnification, low numerical aperture lens? This seems to be obvious from the large scale bar used.
- How was digital image analysis done? Were the datasets filtered? What threshold was applied?
Ideally in such a biofilm with lots of matrix, in this case meaning EPS/polysaccharides, it would be a good idea to examine for glycoconjugates by using lectins.
Page 17, lines 676 ff
I guess the confocal instrument was equipped with an inverted microscope, what objective lens was used? What step size was applied to record the 3d data sets?
Author Response
We would like to express our gratitude to the two Reviewers for their comments concerning our manuscript entitled “Exopolysaccharide carbohydrate structure and biofilm formation by Rhizobium leguminosarum bv. trifolii strains inhabiting nodules of Trifolium repens growing on an old Zn-Pb-Cd polluted waste heap area” (IJMS-1121998). We found these comments valuable and helpful in revising our paper. All remarks of the Reviewers have been addressed in the revised version of the manuscript as well as in the list of detailed responses given below. The changes introduced into the text are performed using the ‘track changes’ mode.
Answers to Reviewer #1 comments
- “Biofilm development in 96 well plates – this technique is used by several scientists in the field, however its always the question whether this is sedimentation and sticking of cells/aggregates to a surface OR is it real biofilm development and growth, furthermore biofilms in many cases are growing under some kind of flow and shear force – Nevertheless, in this case we may look into growth somehow as the plates were incubated for 3 days”.
Indeed, this is a true that analysis of biofilms (in 96-well polystyrene MicroWell plate) using Confocal Laser Scanning Microscopy (CLSM) could raise a question whether the images we observed are a result of bacterial cells sedimentation/sticking or, as is commonly accepted, it is a real biofilm formed by bacteria. As the bacterial cultures in polystyrene wells were incubated for 4 days, and this time was sufficient for formation of a biofilm by microorganisms, as was also found in the other studies (e.g., Janczarek et al., 2015), we can expect that the R. leguminosarum bv. trifolii strains are able to form biofilms, which was documented by the crystal violet-stained microplate assay and also by Confocal Laser Scanning Microscopy (CLSM).
- “Live/Dead stain – Similarly to the 96 well issue, the L/D staining kit is judged critically, there are believers and others saying it is nonsense – However, I assume that the authors applied it properly in order to get reliable results. Usually with pure cultures the results are more meaningful.”
We agree with the Reviewer that some scientists have objections against the Live/Dead staining kit. This indeed might be correct, as some scientists say, when a Live/Dead staining kit is used to study multispecies’ biofilms (e.g., Netuschil et al., 2014). Keeping in mind that during the experiment we followed strictly the procedure i.a., we avoided the staining of planktonic cells and we used pure cultures, we consider this technique as suitable and informative.
- “It’s not clear at all why the 96 well grown biofilms were first dried before they were stained with L/D? The same question comes up if we look into confocal microscopy? The big advantage of this 3d technique lies in the fact that biological samples can be examined in the natural hydrated state!”
We are grateful to the Reviewer for drawing our attention to this. We verified carefully the protocol and our procedure, and it is clear that after incubation of the bacterial culture in 96-wells for 4 days at 28 °C under static conditions, discarding the culture supernatant and gentle washing with 0.85% NaCl to remove planktonic cells, the biofilm was not air dried before L/D staining. Therefore, we adapted this in the text (line 683).
The questions of Reviewer 1
- “The confocal datasets are very poor in quality! What are we looking at? Maximum intensity projections? There is no biofilm “architecture” visible. It might be better to project the datasets in a different way.”
We intended to present in Figure 2 and 3 the survivability of Rhizobium cells under different metal doses, and incorrectly used the word “architecture” in the caption of these both Figures. In reply to the Reviewer’s suggestion we decided to change the form of presented data. We placed in Figure 2 images of a better quality, which illustrate biofilm structure and survivability of R. leguminosarum bv. trifolii strains originating from the waste heap area and the non-polluted reference site which were non-exposed and exposed to the highest metal doses.
The value that we examined was a maximum intensity projection described as the sum of the intensity of illumination of the pixels in the image, reflecting the integrated optical density of the preparation. (It is a dimensionless coefficient characterizing the absorption of light in an optical medium, defined as the natural logarithm of the inverse of the transmission coefficient of the medium T: D = log (1 / T) = log (I0 / I), where I0, I luminous intensity of the incident beam and the outgoing beam, respectively). This value is calculated from the area occupied by the biofilm structures, minus the background. As a result of this analysis, we obtain the percentage of live and dead bacteria. It is the standard method used to assess the survival rate of bacteria both in suspensions and in biofilms.
- “Why is there such a high background? This may originate from the 30 min drying which will shrink the original 3d biofilm structure into a more condensed mass of cells.”
The high background results, first of all, can arise as a result of the use of polystyrene plates; the glow is created from the light diffused through them, and secondly, there may be a secondary issue concerning the graphic processing of the photos. It can`t be a result of drying the biofilm because, as we mentioned above, we did not air dry the biofilms before L/D staining of bacteria.
- “Why is the resolution so poor? Probably due to a low magnification, low numerical aperture lens? This seems to be obvious from the large scale bar used.”
The poor resolution is due to the problem described above i.e., it is a result of glow of the scattered light. Poor resolution may also be due to a low magnification. A low magnification (200×) was used because of the goal of these pictures concerned with the analysis of live-dead bacterial cells survival.
- “How was digital image analysis done? Were the datasets filtered? What threshold was applied?”
The ratio of live /dead bacteria in the biofilm using the L/D staining was carried out in the ImageJ 1.50i program (Wayne Rasband, National Institutes of Health, USA). All photographs have been calibrated and filtered out from the background. As the photographs were recorded in RGB mode, they were separated for analysis into three basic color channels, subsequently converted to 8-bit grayscale, and then analyzed using the Triangle method to identify the threshold value.
- “I guess the confocal instrument was equipped with an inverted microscope, what objective lens was used? What step size was applied to record the 3d data sets?”
Yes, the images were recorded on the inverted microscope (AxioVert 200M, Zeiss). A Plan-Neofluar 20× 0.50 (Zeiss) objective lens was used for the photographs presenting bacterial survival, while a Plan-Neofluar 40× 0.75 (Zeiss) objective lens was used for the photographs of the planes, which were then used to reconstruct the 3D image. Between the scanned planes were 1-μm step spacing.
Reviewer 2 Report
The presented work examines the long-term effect of exposure to heavy metal pollution on EPS structure and biofilm formation of rhizobia, R. leguminosarum bv. trifolii strains. Interestingly, the study reporting significant differences in Exopolysaccharide (EPS) carbohydrate structure, biofilm thickness, and the ratio of living/dead bacteria in the biofilm were found between strains originating from a heavy metal wasted heap and from the reference site both without exposure to metals and under metal exposure. Finally, the authors suggested that studied rhizobia can be assumed as potentially useful in remediation processes.
The work and results are well presented and discussed in this manuscript. The reviewer has only a few comments as following:
Language:
The language of the manuscript needs a minor revision, a few examples are listed below:
- The authors should be consistent when using the oxford comma. The manuscript has missed this comma in some cases (e.g., P3, L140.)
- The manuscript contains long sentences such as the one on P15, L556-563, please revise the length of sentences through the manuscript.
Results:
- Table 1: The authors presented the quantity of identified carbohydrate composition of EPS in 14 samples in table formate. The table also provides a visual illustration of the differences between samples via “heat-map” coloring. However, the color scale is not adequate to show the differences (e.g., 15.13 and 32.68 have the same color even though they appear to be significantly different). Consider changing the coloring scale.
- Figures: consider improving the resolution of Figure 1, S1, and S2
Author Response
We would like to express our gratitude to the two Reviewers for their comments concerning our manuscript entitled “Exopolysaccharide carbohydrate structure and biofilm formation by Rhizobium leguminosarum bv. trifolii strains inhabiting nodules of Trifolium repens growing on an old Zn-Pb-Cd polluted waste heap area” (IJMS-1121998). We found these comments valuable and helpful in revising our paper. All remarks of the Reviewers have been addressed in the revised version of the manuscript as well as in the list of detailed responses given below. The changes introduced into the text are performed using the ‘track changes’ mode.
Answers to Reviewer #2 comments
- “The authors should be consistent when using the oxford comma. The manuscript has missed this comma in some cases (e.g., P3, L140.)”.
We carefully checked the manuscript and introduced the missing oxford commas.
- “The manuscript contains long sentences such as the one on P15, L556-563, please revise the length of sentences through the manuscript.”
We carefully checked the whole text and shortened several long sentences.
- “Table 1: The authors presented the quantity of identified carbohydrate composition of EPS in 14 samples in table formate. The table also provides a visual illustration of the differences between samples via “heat-map” coloring. However, the color scale is not adequate to show the differences (e.g., 15.13 and 32.68 have the same color even though they appear to be significantly different). Consider changing the coloring scale.”
We adjusted the color scale in the Table 1.
- “Figures: consider improving the resolution of Figure 1, S1, and S2”.
According to the suggestion we improved the resolution of Figure 1, S1, and S2.